# SurGE: A Benchmark and Evaluation Framework for Scientific Survey Generation

## Abstract

The rapid growth of academic literature makes the manual creation of scientific surveys increasingly infeasible. While large language models show promise for automating this process, progress in this area is hindered by the absence of standardized benchmarks and evaluation protocols. To bridge this critical gap, we introduce **SurGE** (Survey Generation Evaluation), a new benchmark for scientific survey generation in computer science. SurGE consists of (1) a collection of test instances, each including a topic description, an expert-written survey, and its full set of cited references, and (2) a large-scale academic corpus of over one million papers. In addition, we propose an automated evaluation framework that measures the quality of generated surveys across four dimensions: comprehensiveness, citation accuracy, structural organization, and content quality. Our evaluation of diverse LLM-based methods demonstrates a significant performance gap, revealing that even advanced agentic frameworks struggle with the complexities of survey generation and highlighting the need for future research in this area.[1].

## 1 Introduction

The volume of scientific literature has been expanding at an unprecedented rate in recent years. For instance, academic archives like arXiv now receive over a thousand new computer science papers daily, more than doubled between 2019 and 2024 (Liang et al., 2025). This exponential growth in publications has made the manual creation of comprehensive survey papers increasingly impractical, as the manual collection and synthesis of large volumes of relevant papers is both labor-intensive and time-consuming. Faced with this challenge, there is a growing need for automated systems that can generate survey papers effectively. While recent advancements in LLM-based agents (Yao et al., 2023; Wang et al., 2024c) offer significant promise for automating this task, their full potential is severely hindered by the lack of reliable, scalable, and standardized evaluation benchmarks. Currently, the evaluation of automatically generated surveys largely depends on human-expert reviews, which limit the reproducibility and objectivity of assessments (Tian et al., 2024a). Consequently, progress in this field is difficult to quantify, and the relative merits of different methods remain challenging to compare without standardized, multi-faceted evaluation benchmarks.

To address these challenges and fill this research gap, we introduce **SurGE** (Survey Generation Evaluation), a novel benchmark that establishes a reproducible standard for the survey generation task, which we formalize as a two-stage process: (1) retrieving relevant papers for a given topic from a large corpus, and (2) synthesizing them into a well-structured survey. To support standardized evaluation of both stages, SurGE provides a comprehensive dataset consisting of two key components. The first is a large-scale academic corpus containing over one million computer science papers to be used as the search and retrieval pool. The second is a collection of test instances, where each instance comprises a research topic (e.g., "Machine Learning for Information Retrieval") and its corresponding ground-truth survey. Each ground-truth survey is a high-impact, peer-reviewed paper, chosen based on strict citation criteria and subsequently validated for its quality by expert annotators (see Section 3.1), and is provided along with its full list of cited references.

---

[1]We have open-sourced all the code, data, and models at: `https://anonymous.4open.science/r/SurGE_Benchmark`

Beyond dataset construction, another key contribution of this work is a fully automated evaluation framework that enables reproducible and multi-faceted assessment of generated surveys. Grounded in established principles for high-quality survey writing (Webster & Watson, 2002; Boote & Beile, 2005; Keele et al., 2007; Pautasso, 2013), our proposed evaluation framework quantifies a survey's quality across four key dimensions: comprehensiveness, citation accuracy, structural organization, and content quality. This automated approach eliminates the reliance on subjective and time-consuming human evaluation, thus establishing a scalable and standardized methodology to guide and measure progress in survey generation.

To validate the utility of SurGE, we evaluate a range of LLM-based baseline systems on the benchmark. These include standard retrieval-augmented generation approaches that first retrieve relevant documents and then generate a survey, as well as more advanced pipelines that incorporate explicit planning (e.g., outline generation) and iterative refinement of the draft. The results reveal that even the state-of-the-art systems struggle with the survey generation task. For example, models often miss important papers, produce fragmented or imbalanced coverage of subtopics, and sometimes generate hallucinations with irrelevant citations. These findings indicate a significant gap between machine-generated surveys and expert-written ones, underscoring the need for further research. We believe SurGE will spur the development of more effective techniques at the intersection of information retrieval and generative modeling to tackle this challenging task.

In summary, our contributions are threefold:

1. We introduce **SurGE**, a comprehensive benchmark for scientific survey generation, featuring expert-written ground-truth surveys and a large-scale academic paper corpus.
2. We propose an automated evaluation framework that assesses survey quality across four crucial dimensions: comprehensiveness, citation accuracy, structure, and content.
3. We provide extensive baseline results and analyses, offering reference points and highlighting key challenges in this emerging task.

## 2 TASK DEFINITION

We formalize scientific survey generation as a two-stage task. Given a topic description $t$ and a large academic corpus $D = \{d_1, d_2, \ldots, d_n\}$, the goal is to automatically generate a survey article $S$ that provides a structured and comprehensive overview of the topic. The process consists of:

- **Document Collection:** A retrieval module or complex agentic retrieval system collects a topic-relevant paper set $\mathcal{R}_t \subseteq D$ containing papers relevant to $t$.
- **Survey Generation:** A generative model composes a well-structured survey $S$ based on the topic $t$ and the retrieved document set $\mathcal{R}_t$, including proper citations and a reference list.

To enable standardized benchmarking of the survey generation task, SurGE offers a comprehensive framework that formalizes the process. The benchmark includes 205 expert-selected topics, each paired with a topic description $t$ and a ground-truth survey for evaluation. SurGE also provides a large-scale academic corpus $D$ containing over one million papers, which supports the document retrieval stage. The generated surveys are then evaluated through an automated system, which quantitatively assesses their quality across four key dimensions. Details of the dataset construction process are outlined in Section 3, while the evaluation framework is explained in Section 4.

## 3 DATASET CONSTRUCTION

### 3.1 GROUND-TRUTH SURVEY COLLECTION AND EXPERT ANNOTATION

To construct the SurGE benchmark, we began by collecting a diverse set of high-quality reference surveys (ground-truth surveys) from recent computer science literature. Candidate texts were drawn from the arXiv repository, focusing on publications between 2020 and 2024 that self-identified as survey articles or systematic reviews. To ensure the academic significance and reliability of each instance, we applied the following selection criteria: (i) the document must explicitly declare itself as a survey or review; (ii) it must have achieved a minimum citation count of 20, indicating scholarly impact (Bornmann & Daniel, 2008); and (iii) the publication date must be between 2020 and 2024.

Following the initial filtering process, we further refine the SurGE dataset through expert annotation. This process aims to assess not only the citation-based impact of each candidate survey but also its quality from the perspective of experienced researchers. To this end, we recruited a team of four computer science Ph.D. students as annotators. Note that annotating a paper did not require a close reading of the entire document. Annotators could typically complete the task within 8–9 minutes per paper on average. Each candidate document was evaluated by two independent annotators along four key dimensions: (i) citation impact, reflecting the scholarly influence of the paper; (ii) content coverage, indicating how comprehensively the survey summarizes the literature within its scope; (iii) structural coherence, assessing the logical organization and clarity of the document's sections; and (iv) citation quality, which examines the relevance, diversity, and traceability of cited works.

Each annotator labeled the document as either "usable" or "not usable." A survey paper was included in the final dataset only if both annotators independently marked it as usable. In cases of disagreement, the paper was discarded to maintain a conservative quality threshold. To ensure fair and motivated participation, annotators were compensated based on working hours at an average rate of 60 CNY per hour, exceeding the local minimum wage in Beijing. Inter-annotator agreement was quantified using Cohen's Kappa, applied to 250 annotated instances. The resulting score of 0.792 indicates substantial agreement and reinforces the reliability of the quality control process. After this filtering stage, we finalized the dataset with 205 rigorously verified survey papers. The final set of 205 ground-truth surveys included in our benchmark is detailed in Appendix G. For each of the 205 selected ground-truth surveys, we provide its complete reference list, which serves as the gold standard for evaluating the comprehensiveness of a generated survey. The detailed methodology for extracting and processing these references is described in Appendix B.

## 3.2 ACADEMIC CORPUS CONSTRUCTION

A crucial component of the **SurGE** benchmark is a large-scale academic corpus designed to serve as the retrieval pool for the document collection stage. Our corpus is built entirely from scholarly metadata obtained from the arXiv repository. To ensure adherence to ethical and legal standards, we exclusively collected metadata and did not include full-text PDFs, a practice permitted by arXiv's Terms of Use, which designates metadata as public domain under the CC0 license (arx, 2025b).

The corpus was constructed through a two-stage process. The initial stage involved seeding the corpus with the references from our 205 ground-truth surveys. We systematically retrieved the arXiv metadata for every cited paper that was publicly accessible. This process revealed that approximately 30% of the references were unavailable, primarily due to publication in closed-access journals or other restricted venues. In the second stage, we expanded the corpus to enhance its comprehensiveness. We queried the official arXiv search API, using keywords and titles from the ground-truth surveys to identify and collect metadata for other topically related papers. This methodology resulted in a final retrieval corpus of 1,086,992 unique papers. For each paper, the corpus provides rich metadata, including its title, authors, abstract, subject categories, publication date, and a direct link to its arXiv page for transparency and verification. To ensure high data quality, all collected metadata underwent a rigorous pre-processing pipeline involving text normalization, formatting removal, and deduplication.

## 3.3 STATISTICS AND ANALYSIS

The resulting SurGE benchmark comprises 205 ground-truth survey papers and a retrieval corpus of 1,086,992 documents. Table 1a presents the key statistics of our curated dataset. To quantitatively analyze the organizational complexity of the surveys, we model the hierarchical section headings (e.g., Section, Subsection) of each paper as a tree structure. Our analysis reveals that these surveys are structurally deep, with an average tree depth of 3.07 and a mean of 42.7 nodes (i.e., distinct sections) per document. This structural complexity presents a significant challenge for hierarchical text generation. Furthermore, the surveys are densely referenced, citing an average of 65.8 papers, which underscores the demand for high-recall information collection. Table 1b details the pre-processed fields for each survey instance, which include not only standard metadata but also the parsed structural tree and the ground-truth list of cited documents, enabling a fine-grained, multi-faceted evaluation of system-generated surveys.

Table 1: Overview of the SurGE Benchmark. (a) Summary statistics of the curated survey dataset and its associated retrieval corpus. (b) Metadata of the pre-processed survey dataset used in SurGE.

(a) Basic Statistics of the SurGE Dataset and Corpus

| Statistic | Number |
|---|---|
| Total Ground Truth Surveys | 205 |
| Average Tree Depth | 3.073 |
| Maximum Tree Depth | 4 |
| Average Number of Tree Nodes | 42.717 |
| Maximum Number of Tree Nodes | 212 |
| Average Citations per Paper | 65.78 |
| Average Citations per Section | 1.577 |
| Corpus Size | 1,086,992 |
| Average Abstract Length (words) | 156.57 |

(b) Fields in the Pre-processed Survey Dataset

| Field | Description |
|---|---|
| SurveyID | A unique identifier for the survey. |
| Authors | List of contributing researchers. |
| Title | The title of the survey paper. |
| Year | The publication year of the survey. |
| Date | The timestamp of publication. |
| Category | Subject classification following the arXiv taxonomy. |
| Abstract | The abstract of the survey paper. |
| Structure | Hierarchical representation of the survey. |
| All_Cites | List of document IDs cited in the survey. |

## 3.4 ETHICAL CONSIDERATIONS AND LICENSING

Our corpus is constructed exclusively from arXiv-provided descriptive metadata (titles, authors, abstracts, identifiers, categories, and license URIs) harvested via the official API. We do not host or redistribute arXiv PDFs or source files. This design complies with arXiv's API Terms of Use, which place descriptive metadata under a CC0 public-domain dedication (arx, 2025b). This design is also consistent with the arXiv Submittal Agreement's CC0 designation for metadata (arx, 2025a).

We have prioritized transparency, reproducibility, and ethical considerations throughout dataset construction. To support open science, we have publicly released the SurGE dataset, accompanying metadata, and all associated processing scripts on our official GitHub repository[2]. The dataset and codebase are distributed under the MIT license, granting researchers and developers unrestricted access and modification rights. Regular updates will ensure continued relevance and alignment with evolving research trends and standards.

## 4 EVALUATION FRAMEWORK

To comprehensively evaluate the quality of automatically generated scientific surveys, we propose a multi-faceted evaluation framework. This framework assesses survey quality across four crucial dimensions: Comprehensiveness, Citation Accuracy, Structural Quality, and Content Quality. Each generated survey is evaluated against an expert-written Ground Truth (GT) survey. The following subsections define the quantitative metrics for each dimension in detail.

### 4.1 COMPREHENSIVENESS

The comprehensiveness of a scientific survey is a critical quality factor, as the omission of key publications can undermine its credibility and value. To quantify this aspect, we evaluate the Recall of a generated survey's references against the ground-truth reference lists. Formally, let $R_{GT}$ be the set of references in an expert-written GT survey and $R_G$ be the set of references in our generated survey. Recall $\mathcal{R}$ is defined as:

$$\mathcal{R} = \frac{|R_{GT} \cap R_G|}{|R_{GT}|}, \tag{1}$$

While the GT reference set is not assumed to be perfectly complete, it serves as the best available proxy for expert consensus on a topic's core literature, given that our GT surveys are highly cited, peer-reviewed publications (detailed in §3.1). We therefore interpret this metric not as a measure of absolute completeness, but as a pragmatic metric for evaluating a system's ability to identify the central body of work validated by the research community.

---

[2]https://anonymous.4open.science/r/SurGE_Benchmark

## 4.2 CITATION ACCURACY

Citation accuracy is another critical aspect of a high-quality survey. Each citation in a survey should be thematically relevant to the overall topic of the survey, and it must be contextually appropriate in terms of both the section and sentence in which it appears. To evaluate this aspect, we introduce the metric of Citation Accuracy, which evaluates each citation in the survey across three levels. First, at the document level, we assess whether a cited paper is thematically relevant to the overall topic of the survey. Second, at the section level, we evaluate whether a citation is placed in a semantically appropriate section of the survey. Finally, at the sentence level, we verify whether a citation supports the specific claim made in the sentence where it is cited.

To automate this evaluation, we employ a Natural Language Inference (NLI) model (`nli-deberta-v3-base`[3]) to assess the relevance of each citation. An NLI model is designed to determine the logical relationship between two text snippets: a **premise** and a **hypothesis**. The model then predicts the relationship between these two components, providing probabilities for the following labels: ENTAILMENT, NEUTRAL, CONTRADICTION. Due to its ability to capture semantic relationships, NLI has become a standard method for evaluating the factual consistency and relevance of text generated by LLMs, rendering it an ideal tool for our task.

For our specific evaluation, we implement this three-level check by framing it as a series of NLI tasks. For each citation $r$ (with title $T_r$ and abstract $A_r$) within the generated survey $S$ (with title $T_S$), we construct a set of premise-hypothesis pairs. The premise is consistently formulated using the content of the cited paper, providing the evidentiary basis for the claim. The hypothesis is specifically tailored to assert relevance at each of the three levels (document, section, and sentence). This formulation is structured as follows:

---

**NLI Task Formulation**

**Premise** (Consistent for all levels): There is a paper. Title: "$T_r$". Abstract: $A_r$.
**Hypotheses** (Tailored for each level of granularity):

- **Document-level:** The paper titled "$T_r$" with the given abstract is thematically relevant to the survey titled: "$T_S$".

- **Section-level:** The paper titled "$T_r$" with the given abstract is relevant to the section: "*Section Title*".

- **Sentence-level:** The paper titled "$T_r$" with the given abstract supports the claim made in the sentence: "*Sentence Text*".

---

The score for each citation unit is calculated via a multi-step process. Let $R$ denote the set of all citation instances in the generated survey. For each citation $r \in R$, we compute a score at each of the three levels: document ($y_d(r)$), section ($y_s(r)$), and sentence ($y_t(r)$). The calculation proceeds as follows. First, we resolve two special cases without querying the NLI model. Any citation $r$ not found in our academic corpus is classified as a hallucination and assigned a score of $y_x(r) = 0$ at all levels $x \in \{d, s, t\}$. Conversely, any citation $r$ that is present in the ground-truth survey's bibliography is assigned a document-level score of $y_d(r) = 1$. For all other cases, the relevance score $y_x(r)$ is determined by the NLI model's output probabilities for the labels ENTAILMENT, NEUTRAL, CONTRADICTION. The probabilities are mapped to a final score for each unit as follows:

$$y_x(r) = \begin{cases} 1, & \text{if Entailment has the highest score;} \\ 0.5, & \text{if Neutral is highest and Entailment is second-highest;} \\ 0, & \text{otherwise.} \end{cases} \quad x \in \{d, s, t\} \quad (2)$$

Finally, we aggregate the individual citation scores to produce three final metrics for the survey: Document-level Accuracy ($R_d$), Section-level Accuracy ($R_s$), and Sentence-level Accuracy ($R_t$). For each level $x \in \{d, s, t\}$, the score $R_x$ is calculated as the mean of the individual citation scores $y_x(r)$ over all citation instances $\mathcal{R}$ in the survey:

$$R_x = \frac{1}{|R|} \sum_{r \in R} y_x(r), \quad x \in \{d, s, t\}, \quad (3)$$

---

[3]https://huggingface.co/cross-encoder/nli-deberta-v3-base

## 4.3 Structural Quality

The logical flow and coherence of a scientific survey are fundamentally determined by its structure. This makes structural quality a critical factor for readability and overall impact. To comprehensively evaluate structural quality, we introduce two complementary metrics that assess the generated outline at both macroscopic and microscopic levels. Our first metric, the **Structure Quality Score (SQS)**, addresses the high-level organization. It holistically assesses the alignment between the generated and ground-truth outlines by comparing their overall structure, semantic coherence, and topical progression. Complementing this, our second metric, **Soft-Heading Recall (SHR)**, provides a fine-grained evaluation of heading alignment. It specifically measures how well the generated headings cover those in the ground-truth based on the similarity of semantic embeddings.

**Structure Quality Score (SQS).** SQS evaluates the overall quality of a generated survey's structure based on the hierarchical list of its section headings. To compute this score, we adopt the LLM-as-a-Judge paradigm, leveraging GPT-4o as the evaluator. Specifically, we provide the LLM with both the generated and ground-truth outlines and prompt it to assign a quality score. To guide the LLM's evaluation, we have carefully designed a detailed instruction prompt that includes a comprehensive scoring rubric on a scale from 0 to 5. The complete prompt is shown in Appendix F.

**Soft-Heading Recall (SHR).** To measure fine-grained alignment, SHR evaluates how well the generated outline covers the specific headings present in the ground-truth outline. Unlike metrics based on exact lexical matching, SHR leverages semantic similarity to robustly handle variations in wording and paraphrasing. Formally, SHR is defined as the soft cardinality overlap between the predicted heading set ($H_P$) and the ground-truth heading set ($H_{GT}$):

$$\text{SHR} = \frac{\mathcal{S}(H_P \cap H_{GT})}{\mathcal{S}(H_{GT})}, \tag{4}$$

where $\mathcal{S}(A)$ denotes the "soft cardinality" of a heading set $A$. Intuitively, this metric counts the number of semantically unique headings in a set. It achieves this by down-weighting redundant headings. Specifically, the contribution of each heading is inversely proportional to its aggregated similarity with all other headings in the set:

$$\mathcal{S}(A) = \sum_{i=1}^{K} \frac{1}{\sum_{j=1}^{K} \text{sim}(A_i, A_j)}. \tag{5}$$

Here, $\text{sim}(A_i, A_j)$ is the cosine similarity between the embeddings of headings $A_i$ and $A_j$. A standard set intersection would be too strict for comparing paraphrased headings. Therefore, we define the soft intersection cardinality using the inclusion-exclusion principle:

$$\mathcal{S}(H_P \cap H_{GT}) = \mathcal{S}(H_P) + \mathcal{S}(H_{GT}) - \mathcal{S}(H_P \cup H_{GT}). \tag{6}$$

The core idea lies in the union term, $\mathcal{S}(H_P \cup H_{GT})$. When computed on the combined heading set, a predicted heading and a similar ground-truth heading mutually reduce the union's soft cardinality. This reduction directly quantifies their semantic overlap, allowing the metric to reward paraphrased matches. A higher SHR score thus indicates better granular alignment.

## 4.4 Content Quality

To assess the content quality of generated scientific surveys, we propose the Content Quality Score (CQS) metric based on the LLM-as-a-Judge paradigm, leveraging GPT-4o to evaluate each section of the survey. The evaluation is based on five criteria: fluency and coherence, logical clarity, avoidance of redundancy, clarity of description, and absence of errors. To guide the LLM's evaluation, we designed a detailed instruction prompt for the LLM, which is provided in the Appendix F. Each section is scored on a scale of 0 to 5, where a higher score reflects superior fluency, logical progression, and clarity. Considering the context length limitations of the LLM, we have it score each survey section by section, and the final score is the average of the scores from all sections.

As supplementary measures, we also compute ROUGE and BLEU scores, which quantify n-gram overlap between the generated and ground-truth surveys. While these are well-established metrics in text generation, their role is secondary in our framework, serving as additional checks for content fidelity rather than a primary assessment method.

## 5 EXPERIMENTAL SETUP

In this section, we detail the implementation of our experiment. Each baseline follows a two-stage pipeline: (1) retrieving a set of potentially relevant papers for a given topic, and (2) organizing and summarizing the retrieved papers to produce a structured survey. For fair comparison, all baselines share the same dense retriever for the first stage. In the following subsections, we first describe the training of the shared Paper Retriever, followed by the baseline selection and implementation details.

### 5.1 PAPER RETRIEVER TRAINING

We employ a dual-encoder architecture for retrieval, initialized with `roberta-base`. The training process leverages the benchmark dataset, where each topic description $t$ (introduced in §2) serves as the query $q$. For a query $q$ and a paper abstract $d$, we construct their input representations by prepending the special token [CLS] and appending [SEP]. Formally, let $X(q) = [\text{CLS}]\, q\, [\text{SEP}]$ and $X(d) = [\text{CLS}]\, d\, [\text{SEP}]$. We then feed these tokens into `roberta-base` to obtain the contextualized embedding of the [CLS] token:

$$\text{Emb}(X) = \text{transformer}_{[\text{CLS}]}(X). \tag{7}$$

The similarity score between the $q$ and the $d$ is computed as the dot product of their embeddings:

$$S(q, d) = \text{Emb}(X(q))^{\top} \cdot \text{Emb}(X(d)). \tag{8}$$

During training, each query $Q$ is paired with the relevant documents $d^+$ from the ground truth paper to form the positive samples, while negative samples $d^- \in N$ are randomly sampled from the corpus. The retriever is optimized via the softmax cross-entropy loss:

$$\mathcal{L}(Q, d^+, N) = -\log \frac{\exp\big(S(Q, d^+)\big)}{\exp\big(S(Q, d^+)\big) + \sum_{d^- \in N} \exp\big(S(Q, d^-)\big)}. \tag{9}$$

This objective encourages the model to assign higher scores to relevant papers while minimizing scores for irrelevant ones. After training, we use the trained retriever to retrieve the top-ranked papers for each query, thereby providing a collection of relevant papers for the subsequent generation stage.

### 5.2 BASELINES

We selected the following three survey generation baseline methods for our experiments. Detailed descriptions of each method are provided in the appendices.

- **Retrieval-Augmented Generation (RAG):** This straightforward approach retrieves relevant papers, summarizes them in chunks, and then merges these summaries to form the final survey. A detailed description is provided in Appendix E.1.
- **AutoSurvey** (Wang et al., 2024d): This method employs a multi-stage, outline-driven pipeline. It first generates a high-level outline from retrieved papers and then iteratively expands and refines each section. Further details can be found in Appendix E.2.
- **StepSurvey** (Lai et al., 2024b): This approach adopts a granular, step-by-step process. It starts by generating a title and primary headings, which then guide the incremental drafting of each subsection. The methodology is described in detail in Appendix E.3.

### 5.3 IMPLEMENTATION DETAILS.

For the training of Paper Retriever, we randomly split the dataset into a training set and a test set at a ratio of 4:1. We adopt the AdamW optimizer for model optimization, the learning rate is set to $5 \times 10^{-6}$, and the epoch is set to 10. During the training process, we adopt mixed-precision (fp16) training. At inference time, each query retrieves the top 100 relevant papers according to the similarity score. The retriever is initialized using the pre-trained RoBERTa model (Liu et al., 2019).

To ensure a fair comparison, we utilize the Qwen2.5-14B-Instruct model (Yang et al., 2024a) as the base LLM for all baseline methods. For the generation configuration of LLMs, all experiments are conducted using the publicly available implementations provided by Hugging Face. We utilize the

Table 2: Comparison of retrieval models on recalling ground-truth cited papers. The metric is Recall@k, where $k$ is the number of top documents retrieved. Best results are in bold.

| Model | Recall@20 | Recall@30 | Recall@100 | Recall@200 | Recall@500 | Recall@1000 |
|---|---|---|---|---|---|---|
| BM25 | 0.0548 | 0.0652 | 0.1193 | 0.1596 | 0.2213 | 0.2715 |
| Paper Retriever | **0.1706** | **0.2145** | **0.3665** | **0.4681** | **0.6011** | **0.6805** |

Table 3: Main experimental results comparing different survey generation baselines across four dimensions: Comprehensiveness (Comp.), Citation Accuracy, Structural Quality, and Content Quality. Metrics include Recall, Document/Section/Sentence-level Citation Accuracy (Doc-Acc, Sec-Acc, Sent-Acc), Structure Quality Score (SQS), Soft-Heading Recall (SHR), ROUGE-L, BLEU, and Content Quality Score (CQS). The best results are in **bold**, and the second-best results are in underlined.

| Baseline | Comp. | Citation Accuracy | | | Structural Quality | | Content Quality | | |
|---|---|---|---|---|---|---|---|---|---|
| | Recall | Doc-Acc | Sec-Acc | Sent-Acc | SQS | SHR | R-L | BLEU | CQS |
| RAG | 0.0214 | 0.2857 | 0.2502 | 0.2500 | 0.6829 | 0.7900 | 0.1519 | 10.38 | 4.6723 |
| AutoSurvey | 0.0351 | 0.3617 | **0.4935** | **0.4870** | **1.3902** | 0.9697 | 0.1578 | 10.44 | 4.7390 |
| StepSruvey | **0.0630** | **0.4576** | 0.4571 | 0.4636 | 1.1951 | **0.9763** | **0.1590** | **12.02** | **4.8451** |

default hyperparameters and the chat template as outlined in the official Hugging Face repository [4]. All experiments are conducted on a GPU server with 1TB RAM and eight NVIDIA A100 GPUs, each with 40GB of memory.

# 6 EXPERIMENTAL RESULTS

This section presents the detailed evaluation of the selected survey generation baselines. We first evaluate the performance of the shared Paper Retriever (§6.1). This analysis is crucial as it establishes the theoretical upper-bound performance for the comprehensiveness metric and highlights the challenges posed by the initial retrieval stage. Following this, we assess the complete end-to-end performance of the three baselines (§6.2) based on our proposed evaluation framework.

## 6.1 ANALYSIS OF RETRIEVAL PERFORMANCE

A crucial question in our two-stage pipeline is whether performance limitations stem from the retriever's inability to find relevant papers or the generator's inability to use them. To disentangle these factors and quantify the retrieval bottleneck, we evaluate the performance of our fine-tuned dense retriever in isolation. This analysis establishes the theoretical upper bound for the reference coverage that our end-to-end systems can achieve.

We compare our dense retriever against the lexical baseline BM25 (Robertson et al., 2009), using Recall@k as the evaluation metric. This metric measures the percentage of ground-truth papers from the reference survey that are present in the top-k retrieved documents. As shown in Table 2, our fine-tuned Paper Retriever substantially outperforms the BM25 baseline across all values of $k$. The performance gap underscores the inadequacy of lexical search, which struggles to find semantically relevant papers that may not share overlapping keywords. On the other hand, our dense retriever can capture deeper semantic relationships, thus it is far more effective than BM25. However, the results also reveal a critical bottleneck in the survey generation pipeline. Even when retrieving the top 1000 documents (k=1000), Paper Retriever's recall reaches only 68.05% of the ground-truth papers. This performance ceiling imposes a hard upper bound on the downstream generator, as it cannot synthesize information from papers it never receives. Consequently, this performance ceiling underscores that the survey generation task needs more sophisticated retrieval paradigms, such as employing search agents powered by large language models (Zhang et al., 2024).

---

[4]https://huggingface.co/Qwen/Qwen2.5-14B-Instruct

## 6.2 OVERALL PERFORMANCE OF SURVEY GENERATION BASELINES

Our evaluation now transitions from retrieval performance to the end-to-end survey generation task. While the retriever supplies the models with a substantial portion (36.65%) of the ground-truth references, a detailed analysis of the final surveys reveals a significant bottleneck in the generation stage. Table 3 presents a multi-faceted comparison of our three baselines across the four dimensions of survey quality defined by our benchmark. This comprehensive evaluation yields several key insights:

**(1) The Generation Stage is the Primary Bottleneck.** A striking finding is the dramatic performance drop-off between paper retrieval and final survey synthesis. Despite the retriever providing access to 36.65% of ground-truth references (Recall@100), the best-performing model, StepSurvey, incorporates only 6.30% of them into its final output. The standard RAG baseline fares even worse, with a final recall of just 2.14%. This stark disparity highlights that the core challenge lies not in finding relevant papers, but in the generator's ability to effectively identify and integrate crucial information from the retrieved set. Nevertheless, the results also affirm the value of structured pipelines, as AutoSurvey (3.51%) and StepSurvey (6.30%) significantly outperform the naive RAG approach in comprehensiveness.

**(2) AutoSurvey Excels in Local Coherence and Structural Quality.** AutoSurvey demonstrates particular strength in producing locally coherent and well-organized content. As shown in Table 3, it achieves the highest scores for both Section-level Citation Accuracy (Sec-Acc: 0.4935) and Sentence-level Citation Accuracy (Sent-Acc: 0.4870). This suggests its iterative, section-by-section refinement process is highly effective at placing citations within their correct immediate context. Furthermore, their planning approach results in the highest Structure Quality Score (SQS) of 1.3902, indicating superior hierarchical organization in the generated survey.

**(3) StepSurvey Achieves Superior Coverage and Global Content Quality.** In contrast, StepSurvey's strengths lie in metrics related to holistic coverage and overall content quality. It attains the highest **Recall** (0.0630) and **Document-level Citation Accuracy** (Doc-Acc: 0.4576), demonstrating a better ability to cover the breadth of the topic and align cited works with the survey's main theme. Its multi-phase workflow that generates headings, then subtopics, and finally assigns citations leads to the best content quality. This is evidenced by its top scores in text similarity metrics (**ROUGE-L**: 0.1590, **BLEU**: 12.02) as well as the highest **Content Quality Score (CQS)** of 4.8451, which reflects stronger logical flow and presentation.

**(4) A Trade-off Between Local Precision and Global Coverage.** The distinct strengths of AutoSurvey and StepSurvey reveal a fundamental trade-off in planning strategies. AutoSurvey's iterative refinement of individual sections fosters high-quality local structure and precise, fine-grained citation placement. Conversely, StepSurvey's hierarchical, topic-first approach yields better overall topic coverage and more coherent, globally relevant content. While both advanced methods significantly outperform the standard RAG pipeline, neither excels across all dimensions.

In conclusion, our proposed SurGE benchmark effectively diagnoses the weaknesses of current survey generation systems. The results clearly indicate that advanced, multi-stage planning is crucial, but a substantial gap remains between machine-generated and expert-written surveys, particularly in comprehensiveness. The observed trade-off between local and global optimization strategies highlights a key challenge for future research: developing hybrid models that can combine the fine-grained accuracy of AutoSurvey with the broad thematic coverage and coherence of StepSurvey.

## 7 CONCLUSION

In this paper, we introduce **SurGE**, a comprehensive benchmark designed to address the critical need for standardized and reproducible evaluation in automated scientific survey generation. SurGE provides a large-scale academic corpus, a set of expert-written ground-truth surveys, and a fully automated framework to evaluate surveys on comprehensiveness, citation accuracy, structure, and content. Our experiments reveal significant limitations in state-of-the-art LLM-based systems, highlighting challenges such as incomplete topic coverage and reference hallucination. We believe SurGE will catalyze future research and guide the development of more effective LLM-based systems at the intersection of information retrieval and generative AI for this important task.

## 8 ETHICS STATEMENT

The SurGE benchmark aims to facilitate the advancement of automated survey generation and evaluation in the context of scientific research. By utilizing publicly available metadata from arXiv, our work prioritizes transparency and open access. We do not host or redistribute arXiv's PDF content, ensuring compliance with arXiv's Terms of Use and the CC0 public-domain dedication for metadata. While the use of large-scale academic data in SurGE poses minimal ethical risks, we emphasize the importance of ethical data usage, privacy, and responsible development of AI systems. We encourage the research community to consider these aspects when applying or building upon SurGE.

## 9 REPRODUCIBILITY STATEMENT

To ensure reproducibility, all code, data, and models for the SurGE benchmark are open-sourced and available at our official GitHub Repository[5]. Detailed descriptions of the dataset construction, processing steps, and evaluation methods can be found at our official GitHub Repository and appendices. Our open-access approach is designed to facilitate the replication of results and encourage future research in the field of survey generation using large language models.

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

# A  RELATED WORK

## A.1  RETRIEVAL-AUGMENTED GENERATION

Large Language Models (LLMs) are inherently limited by their static, pre-trained parametric knowledge. To address these limitations, Retrieval-Augmented Generation (RAG) has emerged as a key paradigm (Gao et al., 2023; Lewis et al., 2020; Dong et al., 2025; Tu et al., 2025; Su et al., 2025a). By grounding the model in external knowledge, RAG directly addresses several fundamental limitations of LLMs, offering a robust mechanism to mitigate hallucinations (Ji et al., 2023; Su et al., 2024d;b; Wang et al., 2025a), facilitate knowledge updating (Fang et al., 2024a; Wang et al., 2024b;a; 2025b), and enable effective domain adaptation (Yang et al., 2024b; Su et al., 2025c; 2024a;e).

The conventional approach to traditional RAG is built upon the "Retrieval-then-Read" paradigm (Borgeaud et al., 2022; Guu et al., 2020; Lewis et al., 2020). Within this framework, a user's query triggers a search module for relevant documents within a large-scale external corpus. This retrieval step is carried out by either an external retriever (Zhai, 2008; Su et al., 2023b; Robertson et al., 2009; Su et al., 2023a; Ma et al., 2023; Fang et al., 2024b) or a more sophisticated retrieval system (Su et al., 2023c; Salemi & Zamani, 2024). Building upon this foundation, recent work has proposed more advanced RAG architectures to improve efficiency and effectiveness. For instance, Dynamic RAG (Jiang et al., 2022; Su et al., 2024c; Yao et al., 2024) moves beyond a single retrieval step by adaptively triggering the retriever during generation, specifically when the LLM is uncertain during the generation process. From another angle, GraphRAG (Edge et al., 2024) enhances the knowledge source by querying pre-constructed knowledge graphs instead of unstructured text, allowing it to retrieve interconnected facts and relationships. Furthermore, the Parametric RAG paradigm (Su et al., 2025b; Tan et al., 2025; Fleshman & Van Durme, 2025) alters the knowledge injection step by directly injecting retrieved knowledge into the LLM's parameters.

The scientific survey generation task, which is the focus of our SurGE benchmark, presents a significant challenge for even these advanced RAG systems. Unlike typical question-answering, survey generation demands the synthesis of a large, diverse set of documents into a coherent, well-structured survey paper. Therefore, while RAG provides the foundational technology, our SurGE benchmark is specifically designed to push the boundaries of current models by rigorously evaluating their capabilities in large-scale multi-document synthesis and structured content creation.

## A.2  LONG-FORM TEXT GENERATION AND EVALUATION

Long-form text generation is substantially more challenging than short-text generation due to its inherent requirements for sustained coherence and rich contextual understanding. Early approaches mainly used generative adversarial networks and reinforcement learning to conduct long-sequence generation (Guo et al., 2018). More recently, large language models have emerged as a strong tool for this task, offering advanced capabilities to handle long-text generation. For example, structured planning techniques and specialized inference mechanisms are proposed (Sloan et al., 2024; Jin et al., 2024) to generate consistent and high-quality clinical reports. Similarly, hierarchical planning frameworks have demonstrated that content control and multi-constraint instruction following can significantly enhance logical flow and topic coverage (Hu et al., 2022; Pham et al., 2024). Beyond medical or other task-specific applications, context-driven retrieval strategies, such as tree-structured retrieval, can support open-domain long-text generation by guiding the model through extensive knowledge sources (Roy et al., 2024). The effective evaluation framework is vital for measuring the quality, factualness, and user-centric utility of the long-form text generation task. Traditional metrics, designed for shorter texts, often fail to capture the intricacies of longer outputs. Recent work has introduced task-focused benchmarks that emphasize user-oriented objectives, such as personalized writing or domain-specific content generation (Kumar et al., 2024; Salemi et al., 2025). In parallel, factuality assessment has attracted growing interest, with methods proposed to evaluate both verifiable and unverifiable claims. Metrics such as VERISCORE and FACTSCORE break down generated text into atomic facts, checking each for consistency against reliable sources (Song et al., 2024; Min et al., 2023). Beyond factual correctness, coherence and structural quality have been studied extensively. Benchmarks like LongGenBench and HelloBench underscore the importance of evaluating a model's ability to maintain logical organization and clarity over extended passages (Wu et al., 2025; Que et al., 2024).

### A.3 SURVEY GENERATION

In the domain of scientific writing, survey generation involves distilling extensive textual resources into a coherent and structured overview. Recent advances in AI-assisted systems have provided prompting-based approaches to expedite the drafting process while preserving content accuracy (Gero et al., 2022; Kacena et al., 2024). One of the most commonly used approaches is retrieval-augmented generation, which combines large-scale knowledge retrieved from documents with language generation empowered by LLMs to yield factually comprehensive overviews (Lewis et al., 2021). Retrieval-augmented generation is often initiated with dense retrieval methods based on dual-encoder architectures to identify highly relevant documents (Karpukhin et al., 2020). Once these documents are retrieved, summarization techniques—spanning top-down, bottom-up, and graph-based ranking methods—play a pivotal role in producing concise yet faithful summaries (Nayeem & Rafiei, 2024; Pang et al., 2022; Bleiweiss, 2023). Building on these retrieval and summarization-based methodologies, automated literature survey generation has garnered increasing attention (Tian et al., 2024a; Lai et al., 2024a). However, existing techniques depend on limited ground truths and employ coarse evaluation metrics, resulting in oversimplified assessments of survey quality (Wang et al., 2024d). To address these challenges, we present a refined ground truth and a multi-dimensional evaluation framework that emphasizes both accuracy and structural coherence. By evaluating quality through multiple dimensions, our proposed framework advances the capabilities of automated survey generation, offering a more comprehensive and rigorous approach to summarizing scientific literature.

## B REFERENCE EXTRACTION AND PROCESSING

For each of our selected ground-truth surveys, we extracted reference data from its LaTeX source and the associated BibTeX file. First, we parsed the LaTeX source files using custom regular expressions to extract all citation keys (e.g., `\cite{...}` commands). Next, we used these keys to look up the corresponding entries in the BibTeX files and retrieve their complete metadata, including titles, authors, and publication years. These metadata served as unique digital identifiers for each reference. Finally, to ensure data quality, we performed a cleaning step where we systematically removed duplicates and filtered out entries with inconsistencies, such as malformed or excessively long titles.

The core reason for extracting the full reference list from each ground-truth survey is to create a gold standard for evaluating the information collection stage. We operate on the premise that the citations in an expert-written survey represent a curated collection of the field's most foundational and relevant literature. By using this set as the ground truth, we can quantitatively measure the coverage and recall of a system-generated survey's references, providing a clear metric for its performance.

## C DETAILS OF THE ACADEMIC CORPUS

Table 4 provides an overview of the metadata schema used in our academic corpus. Each entry is structured to support efficient retrieval and interpretability.

Table 4: Fields and Descriptions for the Literature Knowledge Base.

| Key | Description |
|---|---|
| Title | The title of the research paper. |
| Authors | A list of contributing researchers. |
| Year | The publication year of the paper. |
| Date | The exact timestamp of the paper's release. |
| Abstract | The abstract of the paper. |
| Category | The subject classification following the arXiv taxonomy. |
| doc_id | A unique identifier assigned for reference and retrieval. |

## D  THE USE OF LARGE LANGUAGE MODELS

Large language models were used as a tool to help refine the language and improve the clarity of this manuscript. Following any such use, the authors reviewed and edited the content to ensure its accuracy and originality, and take full responsibility for the final text.

## E  BASELINE IMPLEMENTATION

### E.1  RETRIEVAL-AUGMENTED GENERATION

In the first baseline, we combine retrieval with a direct generation approach. Given a topic $t$, we use the above retriever to collect the top 100 candidate papers. To manage lengthy inputs, these retrieved papers are split into smaller groups, each containing an approximately equal number of references. We then prompt a large language model (LLM) to summarize each group separately, guiding it to preserve references to the original papers. Formally, for each group of papers $G_k = \{d_1, d_2, \ldots, d_{n_k}\}$, the LLM is conditioned on the sequence of paper abstracts and instructed to produce a partial summary $\hat{S}_k$. Finally, we merge the partial summaries $\hat{S}_1, \hat{S}_2, \ldots$ into a unified survey. This merging step is performed by prompting the LLM once again with all partial summaries and asking for an integrated, logically coherent survey. Although this baseline follows a straightforward two-step approach, it provides a clear assessment of how effective retrieval-based summarization can be when coupled with an LLM's generative capabilities.

### E.2  AUTOSURVEY

AutoSurvey (Wang et al., 2024d) implements a multi-stage survey generation pipeline that starts with a high-level outline and proceeds through iterative expansions. We adapt it to use our fine-tuned retriever in place of its original retrieval mechanism and keep the number of retrieved references consistent for fairness. In the adapted workflow, we first issue a query based on the topic $t$ to retrieve an initial collection of papers $P_{\text{init}}$. The LLM is then prompted to create a structured outline, which includes main sections and subsections tailored to the subject matter. Next, each section is expanded by conditioning on the subset of papers most relevant to that specific section, producing a draft that includes references in a bracketed format (e.g., "[id]"). Once each section is drafted, the LLM refines it to address factual inconsistencies, stylistic mismatches, and reference-formatting issues. Finally, all refined sections are merged into a coherent final survey, with transitions and citation references carefully aligned. The workflow iterates over these stages, leading to incremental improvements in both thematic coverage and presentation quality.

### E.3  STEPSURVEY

StepSurvey (Lai et al., 2024b) is a more granular generation strategy that also begins with retrieving the top 100 candidate papers for a given topic $t$ but proceeds through distinct planning and drafting phases in a sequential manner. This baseline is proposed by a team named "ID" in the NLPCC2024 competition task 6 (Tian et al., 2024b). Rather than producing an overarching outline at once, it starts by proposing a survey title and a set of primary headings that collectively capture the central themes of the retrieved literature. Subsequently, it uses the primary headings to guide the selection of secondary or finer-grained topics, each mapped to a relevant subset of the retrieved papers. The LLM then produces a full draft by writing each subsection with explicit attention to references and academic conventions, thereby encouraging greater control and consistency across sections. Throughout this process, the system attempts to maintain a balanced level of detail, striving for a clear exposition of important subtopics while avoiding excessive verbosity or redundancy. By structuring the content in incremental steps, StepSurvey aims to achieve coherent organization and thorough coverage of the literature.

# F  PROMPT TEMPLATE

This section provides the detailed prompt templates used for our LLM-as-a-Judge evaluation, as described in the main body of the paper. These prompts are specifically designed for GPT-4o to assess the Structure Quality Score (SQS) and the Content Quality of the generated surveys.

---

**Prompt Template for Structure Quality Score**

You are an AI evaluator. Your task is to compare the generated titles with the target titles and assign a score from 0 to 5 based on their similarity in structure, meaning, and wording.

**Target Titles:** {Ground Truth Titles}

**Generated Titles:** {Generated Titles}

**Scoring Criteria:**

**5 – Almost Identical**:
- Nearly all key words match exactly.
- The meaning is fully preserved.
- The phrasing and structure are identical or differ only in trivial ways.

**4 – Very Similar**:
- Most key words match.
- The meaning is nearly identical.
- The phrasing and structure are very close, with minor rewording.

**3 – Similar**:
- Several key words are shared.
- The meaning is largely the same with slight variations.
- The structure is somewhat similar, but there may be word substitutions.

**2 – Somewhat Similar**:
- Some key words are shared, but others are different.
- The general topic is the same, but the emphasis may differ.
- The sentence structures are different but not entirely unrelated.

**1 – Somewhat Different**:
- Few words overlap, but they are not key terms.
- The meaning is somewhat related but mostly different.
- The sentence structures are significantly different.

**0 – Completely Different**:
- Nearly no words in common.
- Completely different meanings.
- No similarity in structure or phrasing.

**Instruction:** Analyze the generated titles based on the criteria above and provide a single score between 0 and 5.

---

## Prompt Template for Content Quality Score

You are an advanced AI language evaluator. Your task is to assess the logical coherence and clarity of the text based on the following criteria:

1. **Fluency and Coherence** – Does the text flow naturally? Are the sentences well-connected and easy to read?

2. **Logical Clarity** – Is the reasoning clear and structured? Does the argument progress logically without contradictions?

3. **Avoidance of Redundancy** – Does the text avoid unnecessary repetition?

4. **Clarity of Description** – Are ideas, concepts, or events described in a way that is easy to understand?

5. **Absence of Errors** – Does the text contain grammatical mistakes, spelling errors, or factual inconsistencies?

You will provide a **score from 0 to 5** based on the following criteria:

**5 – Excellent**:
A score of 5 is awarded to texts that are highly fluent, featuring smooth transitions and a natural flow. The logical progression is clear, well-structured, and easy to follow. There is no redundancy; each sentence contributes meaningfully to the overall message. Furthermore, descriptions are precise and unambiguous, and the text is free of any spelling, grammatical, or factual errors.

**4 – Good**
A score of 4 indicates a text that is mostly fluent but may have minor awkward transitions. Its logical progression is clear, though it might contain slight inconsistencies. There may be some minor redundancy or repetition present. While descriptions are mostly clear, they could contain minor ambiguities, and the text has very few spelling or grammatical errors.

**3 – Average**
A score of 3 applies to texts that are understandable despite containing noticeable awkward phrasing. The logical flow is inconsistent, which may cause some points to feel out of place. Some redundancy or repetition is present and slightly affects readability. Additionally, certain descriptions are vague or unclear, and the text contains some spelling or grammatical mistakes but remains readable.

**2 – Poor**
A score of 2 is assigned when a text is difficult to read due to an awkward structure and poor fluency. Logical inconsistencies make the central argument unclear, and repetitive phrases render the content tedious. Descriptions are vague, making it hard to understand key points, and the submission is characterized by multiple grammatical and spelling errors.

**1 – Very Poor**
A score of 1 denotes a text that is highly disjointed, making it very hard to read. The logical flow is almost nonexistent, with abrupt topic shifts throughout. Redundant sentences are included but add no value to the content. Descriptions are confusing or overly vague, and the text suffers from frequent spelling and grammatical mistakes.

**0 – Incoherent**
A score of 0 is reserved for a text that is completely nonsensical or unreadable. It demonstrates no logical progression or coherence. The content exhibits extreme redundancy or devolves into word salad, and severe errors throughout make it impossible to understand any intended meaning.

**Instruction**

Now evaluate the following paragraph based on the criteria above and provide a score from 0 to 5.

**Paragraph:**

{Paragraph}

# G  SELECTED GROUND TRUTH SURVEY

The following three tables provide the complete list of the ground-truth surveys in the SurGE benchmark. For each entry, we list the title, publication year, primary arXiv category, and citation count. The provided citation counts represent a snapshot from Google Scholar on May 10, 2025.

| Survey Title | Year | Category | Citation Count |
|---|---|---|---|
| A Survey on Edge Computing Systems and Tools | 2019 | cs.DC | 384 |
| A Survey on Graph-Based Deep Learning for Computational Histopathology | 2021 | cs.LG | 141 |
| A Survey of Uncertainty in Deep Neural Networks | 2021 | cs.LG | 1547 |
| A Survey on Explainability in Machine Reading Comprehension | 2020 | cs.CL | 50 |
| MAC Protocols for Terahertz Communication: A Comprehensive Survey | 2019 | cs.NI | 154 |
| Event Prediction in the Big Data Era: A Systematic Survey | 2020 | cs.AI | 175 |
| A Survey on Deep Neural Network Compression: Challenges, Overview, and Solutions | 2020 | cs.LG | 142 |
| Analysis of the hands in egocentric vision: A survey | 2019 | cs.CV | 113 |
| Neural Machine Translation for Low-Resource Languages: A Survey | 2021 | cs.CL | 321 |
| Physics-Guided Deep Learning for Dynamical Systems: A Survey | 2021 | cs.LG | 115 |
| A Survey on Bias and Fairness in Machine Learning | 2019 | cs.LG | 6292 |
| Generative Adversarial Networks for Spatio-Temporal Data: A Survey | 2020 | cs.LG | 136 |
| Ubiquitous Acoustic Sensing on Commodity IoT Devices: A Survey | 2019 | cs.SD | 82 |
| A Survey of Black-Box Adversarial Attacks on Computer Vision Models | 2019 | cs.LG | 120 |
| A Comprehensive Survey on Pretrained Foundation Models: A History from BERT to ChatGPT | 2023 | cs.AI | 726 |
| A Survey on Dynamic Network Embedding | 2020 | cs.SI | 49 |
| A Survey of Moving Target Defenses forNetwork Security | 2019 | cs.CR | 295 |
| A Survey on the Evolution of Stream Processing Systems | 2020 | cs.DC | 111 |
| Deep Gait Recognition: A Survey | 2021 | cs.CV | 264 |
| Transformers in Vision: A Survey | 2021 | cs.CV | 3276 |
| Image Classification with Deep Learning in the Presence of Noisy Labels: A Survey | 2019 | cs.LG | 433 |
| Change Detection and Notification of Web Pages: A Survey | 2019 | cs.IR | 26 |
| A Survey on Deep Learning-based Architecturesfor Semantic Segmentation on 2D images | 2019 | cs.CV | 276 |
| A Survey on Tiering and Caching in High-Performance Storage Systems | 2019 | cs.AR | 28 |
| Multimodal Learning with Transformers: A Survey | 2022 | cs.CV | 766 |
| Attention, please! A survey of Neural Attention Models in Deep Learning | 2021 | cs.LG | 251 |
| Explanation-Based Human Debugging of NLP Models: A Survey | 2021 | cs.CL | 80 |
| Federated Learning in Mobile Edge Networks: A Comprehensive Survey | 2019 | cs.NI | 2488 |
| Deep Learning for Image Super-resolution:A Survey | 2019 | cs.CV | 2036 |
| Deep Learning for Weakly-Supervised Object Detection and Object Localization: A Survey | 2021 | cs.CV | 25 |
| Survey of Transient Execution Attacks | 2020 | cs.CR | 23 |
| A Survey of Syntactic-Semantic Parsing Based on Constituent and Dependency Structures | 2020 | cs.CL | 47 |
| Efficient Deep Learning: A Survey on Making Deep Learning Models Smaller, Faster, and Better | 2021 | cs.LG | 519 |
| From Statistical Relational to Neurosymbolic Artificial Intelligence: a Survey. | 2021 | cs.AI | 71 |
| Symbolic Logic meets Machine Learning: A Brief Survey in Infinite Domains | 2020 | cs.AI | 50 |
| Context Dependent Semantic Parsing: A Survey | 2020 | cs.CL | 21 |
| A survey of active learning algorithms for supervised remote sensing image classification | 2021 | cs.CV | 651 |
| Generate FAIR Literature Surveys with Scholarly Knowledge Graphs | 2020 | cs.DL | 53 |
| A Survey of Deep Learning for Data Caching in Edge Network | 2020 | cs.NI | 39 |
| Logic Locking at the Frontiers of Machine Learning: A Survey on Developments and Opportunities | 2021 | cs.CR | 29 |
| Affective Computing for Large-Scale Heterogeneous Multimedia Data: A Survey | 2019 | cs.MM | 95 |
| Weakly Supervised Object Localization and Detection: A Survey | 2021 | cs.CV | 348 |
| Compression of Deep Learning Models for Text: A Survey | 2020 | cs.CL | 142 |
| Computer Vision with Deep Learning for Plant Phenotyping in Agriculture: A Survey | 2020 | cs.CV | 85 |
| A Survey on Evolutionary Neural Architecture Search | 2020 | cs.NE | 641 |
| Towards Efficient SynchronousFederated Training: A Survey onSystem Optimization Strategies | 2021 | cs.DC | 34 |
| From Distributed Machine Learning to Federated Learning: A Survey | 2021 | cs.DC | 342 |
| Low-Light Image and Video Enhancement Using Deep Learning: A Survey | 2021 | cs.CV | 522 |
| A Survey of Coded Distributed Computing | 2020 | cs.DC | 28 |
| A Systematic Survey of Regularization and Normalization in GANs | 2020 | cs.LG | 56 |
| Deep Learning for Deepfakes Creation and Detection: A Survey | 2019 | cs.CV | 614 |
| Proximity Perception in Human-Centered Robotics: A Survey on Sensing Systems and Applications | 2021 | cs.RO | 127 |
| A Survey of Transformers | 2021 | cs.LG | 1641 |
| How should my chatbot interact? A survey on social characteristics in human-chatbot interaction design | 2019 | cs.HC | 676 |
| Community detection in node-attributed social networks: a∼survey | 2019 | cs.SI | 324 |
| A Survey of Deep Reinforcement Learning in Recommender Systems: A Systematic Review | 2021 | cs.IR | 73 |
| Learning from Noisy Labels with Deep Neural Networks: A Survey | 2020 | cs.LG | 1380 |
| A Survey on Split Manufacturing: Attacks, Defenses, and Challenges | 2020 | cs.CR | 59 |
| A Survey of Active Learning for Text Classification using Deep Neural Networks | 2020 | cs.CL | 145 |
| A Survey of Knowledge Tracing: Models, Variants, and Applications | 2021 | cs.CY | 51 |
| Dynamic Neural Networks: A Survey | 2021 | cs.CV | 849 |
| Arms Race in Adversarial Malware Detection: A Survey | 2020 | cs.CR | 81 |
| A Comprehensive Survey on Graph Anomaly Detection with Deep Learning | 2021 | cs.LG | 775 |
| Deep Learning for Instance Retrieval: A Survey | 2021 | cs.CV | 292 |
| Deep Learning for Vision-based Prediction: A Survey | 2020 | cs.CV | 59 |

| Survey Title | Year | Category | Citation Count |
|---|---|---|---|
| A survey of face recognition techniques under occlusion | 2020 | cs.CV | 165 |
| Blockchain for 5G and Beyond Networks: A State of the Art Survey | 2019 | cs.NI | 440 |
| A Survey of Knowledge-Enhanced Text Generation | 2020 | cs.CL | 344 |
| Hardware Acceleration of Sparse and Irregular Tensor Computations of ML Models: A Survey | 2020 | cs.AR | 112 |
| Graph Learning for Combinatorial Optimization: A Survey of State-of-the-Art | 2020 | cs.LG | 113 |
| Deep Gaussian Processes: A Survey | 2021 | cs.LG | 39 |
| Graph Learning: A Survey | 2021 | cs.LG | 515 |
| Harnessing the Power of LLMs in Practice: A Survey on ChatGPT and Beyond | 2023 | cs.CL | 981 |
| A Survey on In-context Learning | 2022 | cs.CL | 1788 |
| Centrality Measures in Complex Networks: A Survey | 2020 | cs.SI | 139 |
| A Survey on Adversarial Recommender Systems | 2020 | cs.IR | 244 |
| Towards a Robust Deep Neural Network in Texts: A Survey | 2019 | cs.CL | 90 |
| A Survey of Deep Active Learning | 2020 | cs.LG | 1588 |
| Translation Quality Assessment: A Brief Survey on Manual and Automatic Methods | 2021 | cs.CL | 56 |
| Deep learning for scene recognition from visual data: a survey | 2020 | cs.CV | 23 |
| A Survey of State-of-the-Art on Blockchains: Theories, Modelings, and Tools | 2020 | cs.DC | 170 |
| Computation Offloading and Content Caching Delivery in Vehicular Edge Computing: A Survey | 2019 | cs.NI | 101 |
| Recent Advances in Deep Learning Based Dialogue Systems: A Systematic Survey | 2021 | cs.CL | 322 |
| Explainable reinforcement learning for broad-XAI: a conceptual framework and survey | 2021 | cs.AI | 82 |
| End-to-End Constrained Optimization Learning: A Survey | 2021 | cs.LG | 258 |
| Machine Learning in Generation, Detection, and Mitigation of Cyberattacks in Smart Grid: A Survey | 2020 | cs.CR | 31 |
| Pervasive AI for IoT applications: A Survey on Resource-efficient Distributed Artificial Intelligence | 2021 | cs.DC | 153 |
| A Survey of Deep Learning Approaches for OCR and Document Understanding | 2020 | cs.CL | 84 |
| Taxonomy of Machine Learning Safety: A Survey and Primer | 2021 | cs.LG | 81 |
| Neuron-level Interpretation of Deep NLP Models: A Survey | 2021 | cs.CL | 98 |
| Using Deep Learning to Solve Computer Security Challenges: A Survey | 2019 | cs.CR | 59 |
| Efficient Transformers: A Survey | 2020 | cs.LG | 1500 |
| A Survey of Label-noise Representation Learning: Past, Present and Future | 2020 | cs.LG | 208 |
| A Survey of Constrained Gaussian Process Regression: Approaches and Implementation Challenges | 2020 | cs.LG | 182 |
| Neural Networks for Entity Matching: A Survey | 2020 | cs.DB | 157 |
| A Survey of Orthographic Information in Machine Translation | 2020 | cs.CL | 37 |
| A Survey of Quantum Theory Inspired Approaches to Information Retrieval | 2020 | cs.IR | 42 |
| A Survey of Deep Meta-Learning | 2020 | cs.LG | 459 |
| A Survey on Deep Learning Techniques for Video Anomaly Detection | 2020 | cs.CV | 46 |
| A Survey on Interactive Reinforcement Learning | 2021 | cs.HC | 128 |
| Applications of Auction and Mechanism Design in Edge Computing: A Survey | 2021 | cs.GT | 76 |
| Abduction and Argumentation for Explainable Machine Learning: A Position Survey | 2020 | cs.AI | 19 |
| A Survey of Exploration Methods in Reinforcement Learning | 2021 | cs.LG | 133 |
| Reinforcement Learning based Recommender Systems: A Survey | 2021 | cs.IR | 597 |
| A Survey of Data Augmentation Approaches for NLP | 2021 | cs.CL | 1008 |
| A Survey on Self-supervised Pre-training for Sequential Transfer Learning in Neural Networks | 2020 | cs.LG | 77 |
| A Survey on Theorem Provers in Formal Methods | 2019 | cs.SE | 45 |
| A Survey on Heterogeneous Graph Embedding | 2020 | cs.SI | 452 |
| Asynchronous Federated Learning on Heterogeneous Devices: A Survey | 2021 | cs.DC | 326 |
| A Survey on Data-driven Software Vulnerability Assessment and Prioritization | 2021 | cs.SE | 94 |
| Time Series Data Imputation: A Survey on Deep Learning Approaches | 2020 | cs.LG | 96 |
| A Survey on Low-Resource Neural Machine Translation | 2021 | cs.CL | 59 |
| Benchmark and Survey of Automated Machine Learning Frameworks | 2019 | cs.LG | 523 |
| A Survey on Deep Learning Technique for Video Segmentation | 2021 | cs.CV | 204 |
| Universal Adversarial Perturbations: A Survey | 2020 | cs.CV | 66 |
| A Survey of Machine Learning Methods and Challenges for Windows Malware Classification | 2020 | cs.CR | 73 |
| Serverless Computing: A Survey of Opportunities, Challenges, and Applications | 2019 | cs.NI | 295 |
| A Survey on Subgraph Counting: Concepts, Algorithms and Applications to Network Motifs and Graphlets | 2019 | cs.DS | 218 |
| Survey of Attacks and Defenses on Edge-Deployed Neural Networks | 2019 | cs.CR | 51 |
| Reinforcement learning with human advice: a survey. | 2020 | cs.AI | 86 |
| Domain Generalization: A Survey | 2021 | cs.LG | 1487 |
| Deep Learning for 3D Point Clouds: A Survey | 2019 | cs.CV | 2360 |
| Survey on Causal-based Machine Learning Fairness Notions | 2020 | cs.LG | 109 |
| Towards a Survey on Static and Dynamic Hypergraph Visualizations | 2021 | cs.HC | 38 |
| Graph-based Deep Learning for Communication Networks: A Survey | 2021 | cs.NI | 245 |
| Opportunities and Challenges in Explainable Artificial Intelligence (XAI): A Survey | 2020 | cs.CV | 1062 |
| Domain Adaptation and Multi-Domain Adaptation for Neural Machine Translation: A Survey | 2021 | cs.CL | 108 |

| Survey Title | Year | Category | Citation Count |
|---|---|---|---|
| What Can Knowledge Bring to Machine Learning? – A Survey of Low-shot Learning for Structured Data | 2021 | cs.LG | 28 |
| A Survey of Knowledge Graph Embedding and Their Applications | 2021 | cs.IR | 68 |
| Deep Graph Generators: A Survey | 2020 | cs.LG | 75 |
| Post-hoc Interpretability for Neural NLP: A Survey | 2021 | cs.CL | 309 |
| A Survey on Machine Reading Comprehension: Tasks, Evaluation Metrics and Benchmark Datasets | 2020 | cs.CL | 129 |
| A survey on bias in visual datasets | 2021 | cs.CV | 153 |
| A Comprehensive Survey on Graph Neural Networks | 2019 | cs.LG | 12047 |
| A survey of security and privacy issues in the Internet of Things from the layered context | 2019 | cs.NI | 144 |
| Adversarial Machine Learning in Image Classification: A Survey Towards the Defender's Perspective | 2020 | cs.CV | 217 |
| Survey of Machine Learning Accelerators | 2020 | cs.DC | 230 |
| Online Fair Division: A Survey | 2019 | cs.AI | 55 |
| A Comprehensive Survey of Scene Graphs: Generation and Application | 2021 | cs.CV | 354 |
| A Survey of Numerical Methods Utilizing Mixed Precision Arithmetic | 2020 | cs.MS | 216 |
| Image Data Augmentation for Deep Learning: A Survey | 2022 | cs.CV | 445 |
| Neural Unsupervised Domain Adaptation in NLP—A Survey | 2020 | cs.CL | 357 |
| A Survey of Evaluation Metrics Used for NLG Systems | 2020 | cs.CL | 305 |
| Deep Learning for Micro-expression Recognition: A Survey | 2021 | cs.CV | 99 |
| A Survey of Human-in-the-loop for Machine Learning | 2021 | cs.LG | 735 |
| Deep Fake Detection : Survey of Facial Manipulation Detection Solutions | 2021 | cs.CV | 36 |
| Survey of Hallucination in Natural Language Generation | 2022 | cs.CL | 3976 |
| The Rise and Potential of Large Language Model Based Agents: A Survey | 2023 | cs.AI | 971 |
| Generalized Out-of-Distribution Detection: A Survey | 2021 | cs.CV | 1158 |
| A survey on practical adversarial examples for malware classifiers | 2020 | cs.CR | 23 |
| One-Class Classification: A Survey | 2021 | cs.CV | 196 |
| Survey of Low-Resource Machine Translation | 2021 | cs.CL | 197 |
| A Survey on Visual Transformer | 2020 | cs.CV | 405 |
| Automatic Speech Recognition And Limited Vocabulary: A Survey | 2021 | cs.AI | 61 |
| A Comprehensive Survey on Community Detection with Deep Learning | 2021 | cs.SI | 487 |
| A Survey of Distributed Consensus Protocols for Blockchain Networks | 2019 | cs.CR | 1101 |
| Spatiotemporal Data Mining: A Survey on Challenges and Open Problems | 2021 | cs.LG | 148 |
| RGB-D Salient Object Detection: A Survey | 2020 | cs.CV | 305 |
| A Survey on Embedding Dynamic Graphs | 2021 | cs.LG | 177 |
| Transformers in Time Series: A Survey | 2022 | cs.LG | 1152 |
| Deep Learning for 3D Point Cloud Understanding: A Survey | 2020 | cs.CV | 47 |
| AMMU : A Survey of Transformer-based Biomedical Pretrained Language Models | 2021 | cs.CL | 170 |
| Instruction Tuning for Large Language Models: A Survey | 2023 | cs.CL | 919 |
| A Short Survey of Pre-trained Language Models for Conversational AI-A New Age in NLP | 2021 | cs.CL | 94 |
| A Survey on Evaluation of Large Language Models | 2023 | cs.CL | 2885 |
| D'ya like DAGs? A Survey on Structure Learning and Causal Discovery | 2021 | cs.LG | 364 |
| Machine Learning Testing: Survey, Landscapes and Horizons | 2019 | cs.LG | 1059 |
| Deep Learning-based Spacecraft Relative Navigation Methods: A Survey | 2021 | cs.RO | 100 |
| A Survey on Deep Learning Event Extraction: Approaches and Applications | 2021 | cs.CL | 107 |
| Graph Neural Networks for Natural Language Processing: A Survey | 2021 | cs.CL | 396 |
| Stabilizing Generative Adversarial Networks: A Survey | 2019 | cs.LG | 149 |
| A Survey of Knowledge-based Sequential Decision Making under Uncertainty | 2020 | cs.AI | 26 |
| Automated Machine Learning on Graphs: A Survey | 2021 | cs.LG | 97 |
| Active Divergence with Generative Deep Learning - A Survey and Taxonomy | 2021 | cs.LG | 23 |
| Deep Neural Approaches to Relation Triplets Extraction: A∼Comprehensive∼Survey | 2021 | cs.CL | 63 |
| A Survey on Multi-modal Summarization | 2021 | cs.CL | 73 |
| Generative adversarial networks in time series: A survey and taxonomy | 2021 | cs.LG | 105 |
| Towards Reasoning in Large Language Models: A Survey | 2022 | cs.CL | 841 |
| Survey on reinforcement learning for language processing | 2021 | cs.CL | 180 |
| Creativity and Machine Learning: A Survey | 2021 | cs.LG | 56 |
| A Survey of Deep Reinforcement Learning Algorithms for Motion Planning and Control of Autonomous Vehicles | 2021 | cs.RO | 92 |
| Deep Reinforcement Learning in Computer Vision: A Comprehensive Survey | 2021 | cs.CV | 238 |
| Domain Name System Security and Privacy: A Contemporary Survey | 2020 | cs.NI | 69 |
| Resource Allocation and Service Provisioning in Multi-Agent Cloud Robotics: A Comprehensive Survey | 2021 | cs.RO | 145 |
| Generalizing to Unseen Domains: A Survey on Domain Generalization | 2021 | cs.LG | 1311 |
| Deep Graph Similarity Learning: A Survey | 2019 | cs.LG | 103 |
| Word Embeddings: A Survey | 2019 | cs.CL | 449 |
| Augmented Language Models: a Survey | 2023 | cs.CL | 544 |
| Software Engineering for AI-Based Systems: A Survey | 2021 | cs.SE | 317 |
| Pre-train, Prompt, and Predict: A Systematic Survey of Prompting Methods in Natural Language Processing | 2021 | cs.CL | 5613 |
| A Survey on Knowledge Graph Embeddings with Literals: Which model links better Literal-ly? | 2019 | cs.AI | 97 |
| A comprehensive taxonomy for explainable artificial intelligence | 2021 | cs.LG | 276 |
| Federated Learning Meets Natural Language Processing: A Survey | 2021 | cs.CL | 103 |
| Fairness in Machine Learning: A Survey | 2020 | cs.LG | 1163 |
| Deep Learning for Text Style Transfer: A Survey | 2020 | cs.CL | 318 |
| Retrieval-Augmented Generation for Large Language Models: A Survey | 2023 | cs.CL | 1951 |
| Transformers in Medical Imaging: A Survey | 2022 | eess.IV | 889 |
| A Survey of Privacy Attacks in Machine Learning | 2020 | cs.CR | 368 |
| A Survey on Deep Reinforcement Learning for Data Processing and Analytics | 2021 | cs.LG | 44 |
| A Survey on Dialogue Summarization: Recent Advances and New Frontiers | 2021 | cs.CL | 111 |
| A Survey of Machine Learning for Computer Architecture and Systems | 2021 | cs.LG | 91 |
| Inspect, Understand, Overcome: A Survey of Practical Methods for AI Safety | 2021 | cs.LG | 78 |
| Survey on Evaluation Methods for Dialogue Systems | 2019 | cs.CL | 394 |
| A Survey on Recent Approaches for Natural Language Processing in Low-Resource Scenarios | 2020 | cs.CL | 390 |
| DL-Traff: Survey and Benchmark of Deep Learning Models for Urban Traffic Prediction | 2021 | cs.LG | 180 |

