# OpenReview forum: "SurGE: A Benchmark and Evaluation Framework for Scientific Survey Generation"
_ICLR.cc/2026/Conference — ICLR 2026 Conference Withdrawn Submission_

### Official Review · Reviewer_2P8U · 2025-10-23

**Soundness:** 2
**Presentation:** 2
**Contribution:** 1
**Rating:** 2
**Confidence:** 4

**Summary:**

This paper introduces SurGE，a comprehensive benchmark for scientific survey generation.
1. A curated benchmark dataset consisting of 205 expert-written, high-impact survey papers with their complete reference lists, paired with a retrieval corpus of 1,086,992 papers from arXiv.
2. A comprehensive automated evaluation framework that assesses generated surveys across four dimensions.
3. Baseline experiments evaluating three LLM-based methods (RAG, AutoSurvey, StepSurvey) using Qwen2.5-14B-Instruct.

**Strengths:**

1. This paper constructs a complete automatic review evaluation system, and collects a large number of survey papers by human experts for reference.
2. This paper constructs a large-scale literature corpus.

**Weaknesses:**

1. The contribution of this paper may seem inadequate. How is the dataset constructed in this paper different from other works such as autosurvey [1] and surveyforge [2]?
2. The experiment in this paper is not sufficient. Only qwen-14B is used as LLM for the experiment, and it is not compared with surveyforge, surveyx [3], LLM×MR-V2 [4].


[1] Yidong Wang, Qi Guo, Wenjin Yao, Hongbo Zhang, Xin Zhang, Zhen Wu, Meishan Zhang, Xinyu Dai, Qingsong Wen, Wei Ye, et al. Autosurvey: Large language models can automatically write surveys. Advances in neural information processing systems, 37:115119–115145, 2024.

[2] Xiangchao Yan, Shiyang Feng, Jiakang Yuan, Renqiu Xia, Bin Wang, Bo Zhang, and Lei Bai. Surveyforge: On the outline heuristics, memory-driven generation, and multi-dimensional evaluation for automated survey writing. arXiv preprint arXiv:2503.04629, 2025.

[3] Xun Liang, Jiawei Yang, Yezhaohui Wang, Chen Tang, Zifan Zheng, Shichao Song, Zehao Lin, Yebin Yang, Simin Niu, Hanyu Wang, et al. Surveyx: Academic survey automation via large language models. arXiv preprint arXiv:2502.14776, 2025.

[4] Haoyu Wang, Yujia Fu, Zhu Zhang, Shuo Wang, Zirui Ren, Xiaorong Wang, Zhili Li, Chaoqun He, Bo An, Zhiyuan Liu, et al. Llm × mapreduce-v2: Entropy-driven convolutional test-time scaling for generating long-form articles from extremely long resources. arXiv preprint arXiv:2504.05732, 2025

**Questions:**

1. How is the dataset constructed in this paper different from those of autosurvey [1] and surveyforge [2]?

2. How well do some closed-source models perform? And the performance of other survey generated works [2-4]?

3. The GT of the experiment is 205 real reviews, but most of the years are concentrated before 2022. If the surveys generated by LLM cite updated articles such as 2023-2025, how to deal with these citations that are not in GT, and how to judge whether the LLM citations are accurate?

[1] Yidong Wang, Qi Guo, Wenjin Yao, Hongbo Zhang, Xin Zhang, Zhen Wu, Meishan Zhang, Xinyu Dai, Qingsong Wen, Wei Ye, et al. Autosurvey: Large language models can automatically write surveys. Advances in neural information processing systems, 37:115119–115145, 2024.

[2] Xiangchao Yan, Shiyang Feng, Jiakang Yuan, Renqiu Xia, Bin Wang, Bo Zhang, and Lei Bai. Surveyforge: On the outline heuristics, memory-driven generation, and multi-dimensional evaluation for automated survey writing. arXiv preprint arXiv:2503.04629, 2025.

[3] Xun Liang, Jiawei Yang, Yezhaohui Wang, Chen Tang, Zifan Zheng, Shichao Song, Zehao Lin, Yebin Yang, Simin Niu, Hanyu Wang, et al. Surveyx: Academic survey automation via large language models. arXiv preprint arXiv:2502.14776, 2025.

[4] Haoyu Wang, Yujia Fu, Zhu Zhang, Shuo Wang, Zirui Ren, Xiaorong Wang, Zhili Li, Chaoqun He, Bo An, Zhiyuan Liu, et al. Llm × mapreduce-v2: Entropy-driven convolutional test-time scaling for generating long-form articles from extremely long resources. arXiv preprint arXiv:2504.05732, 2025

---

### Official Review · Reviewer_YUqN · 2025-10-29

**Soundness:** 2
**Presentation:** 2
**Contribution:** 2
**Rating:** 4
**Confidence:** 4

**Summary:**

This paper introduces SurGE, a benchmark for the scientific review generation task. The authors construct a dataset containing 205 high-quality review papers written by experts with complete reference lists, and accompanied by a large-scale retrieval corpus of over one million papers. Further, they propose an automated evaluation framework that assesses the generated reviews from four dimensions: comprehensiveness, citation accuracy, structural quality, and content quality. Experiments show that there is still much room for improvement in the review generation task.

**Strengths:**

The motivation is meaningful - a better evaluation of automatic survey generation could help develop powerful auto-survey pipeline and further improve the efficiency of literature review. Besides, the authors construct a large-scale literature corpus.

**Weaknesses:**

1. The corpus source is too singular, leading to coverage bias: The academic corpus constructed in this paper is entirely from arXiv. This creates an inherent limitation: as the authors state in the paper, 30% of the articles could not be found. Furthermore, the authors only included papers from 2020 to 2024, which fails to cover the latest articles, especially in rapidly developing fields such as large models.

2. The evaluation method overly relies on and lacks reflection on "AI judges": The assessment of "citation accuracy", "structural quality", and "content quality" in the paper heavily depends on other AI models as judges. Although this achieves automation, the paper does not deeply explore or validate the reliability and potential biases of these "AI judges". A human study can possibly verify the effectiveness of this auto-evaluation (the consistency between human eval and llm eval).

3. Some articles about autosurvey are not included in the evaluation such as SurveyForge [A], SurveyX [B].

[A] Yan X, Feng S, Yuan J, et al. Surveyforge: On the outline heuristics, memory-driven generation, and multi-dimensional evaluation for automated survey writing. ACL 2025.

[B] Liang X, Yang J, Wang Y, et al. Surveyx: Academic survey automation via large language models[J]. arXiv preprint arXiv:2502.14776, 2025.

**Questions:**

The authors should further clarify the difference between the evaluation methods proposed in this article and those in autosurvey and surveyforge.

---

### Official Review · Reviewer_ZQ34 · 2025-10-31

**Soundness:** 2
**Presentation:** 3
**Contribution:** 2
**Rating:** 4
**Confidence:** 4

**Summary:**

The authors introduce SurGE, a benchmark and evaluation framework for automated scientific survey generation. It provides 205 ground-truth surveys and a 4-dimension evaluation metric. However, this work suffers from a critical lack of novelty, as its core ideas and evaluation methods are highly derivative of recent work like AutoSurvey and SurveyForge.

**Strengths:**

- The paper correctly identifies a significant need within the community for a standardized, reproducible benchmark for the complex task of survey generation.

**Weaknesses:**

- The idea of a benchmark using human surveys and a multi-dimensional, LLM-based evaluation framework has already been established by SurveyForge and AutoSurvey. SurGE's metrics are derivative and offer no significant conceptual innovation.

- The benchmark's scale is limited, and its construction process lacks novelty. Furthermore, the validation was performed by only "four computer science Ph.D. students." This small number raises concerns about the annotators' breadth of expertise across diverse CS topics and the overall rigor of the ground-truth validation.

**Questions:**

- Ground-truth surveys have a publication cutoff (2020-2024). How does the evaluation framework fairly assess an agent that cites newer, more relevant papers (e.g., from 2025) that are not in the ground-truth reference set? The current comprehensiveness metric seems to fundamentally penalize up-to-date models, which is counter-intuitive for a survey writing task.

- The paper's main contribution appears to be substantially similar to the SurveyBench and SAM evaluation framework proposed in SurveyForge. Could the authors please clearly differentiate their conceptual contribution from this work, beyond minor implementation differences (e.g., Recall vs. Precision for reference overlap)?

---

### Official Review · Reviewer_XfMn · 2025-11-01

**Soundness:** 2
**Presentation:** 3
**Contribution:** 2
**Rating:** 4
**Confidence:** 4

**Summary:**

This paper presents SurGE, a benchmark and automated evaluation framework for scientific survey generation with a focus on computer science literature. SurGE includes (1) a dataset of 205 expert-annotated, high-quality survey papers (with full references and structured metadata), and (2) a one-million-plus document academic corpus for retrieval. The authors introduce a four-dimensional, largely automated evaluation suite (covering comprehensiveness, citation accuracy, structural organization, and content quality) and use it to analyze several LLM-based survey-generation pipelines. The evaluation reveals a significant performance gap between current automated approaches and expert-written surveys, highlighting the challenges of this emerging task.

**Strengths:**

1. Clear Problem Formulation and Task Relevance: The paper identifies the pressing challenge of scalable, objective, and reproducible evaluation for automated scientific survey generation. Positioning SurGE as a two-stage process—retrieval and synthesis—is methodologically sound, and highly relevant given the explosion of literature and the rise of LLM-based knowledge agents.

2. Automated, Multi-dimensional Evaluation: The authors move decisively beyond basic n-gram overlap (e.g., ROUGE, BLEU) by proposing four tailored metrics: recall-based comprehensiveness, citation accuracy via NLI models at multiple granularities, LLM-based structure scoring (SQS and Soft-Heading Recall), and LLM-graded content quality. The mathematical formulation for citation accuracy (see Equations on Page 5) and structure (see Section 4.3) is thoughtful and clear.

**Weaknesses:**

1. Limited Scope of Benchmark Domain and Generalization: SurGE is strictly focused on computer science (specifically, arXiv-sourced material) as both the test domain and the underlying corpus. While this is a pragmatic choice, it raises concerns about the generalizability of the benchmark and framework to other disciplines, such as biomedicine, physics, or social sciences. Although transparency and reproducibility are maintained by using public data, the paper does not provide discussion or empirical evidence about portability to other scholarly domains or tasks.

2. Automated Evaluation Limitations: While the aim of full automation in evaluation is laudable, two of the four main metrics (Structure Quality Score and Content Quality Score) depend on LLM ‘judgement’, specifically GPT-4o, as explained in Section 4.3 and 4.4, and detailed in the corresponding prompt templates in Appendix F. The potential biases, instability, and lack of transparency in LLM graders (especially with evolving Large Language Models) are not sufficiently analyzed or acknowledged; issues such as drift in the LLM prompt, or unexpected outputs from black-box systems, can seriously affect reproducibility/fairness.

3. Human alignment: For tasks requiring high creativity, such as survey writing, the LLM-as-a-judge evaluation paradigm must be rigorously validated against human expert assessment to prove its reliability. The evaluation of creative writing cannot be reduced to a simple checklist or template-based scoring; instead, assessing more subjective aspects of quality is both more challenging and valuable, and the alignment between automated and human judgment in these areas must be explicitly demonstrated.

**Questions:**

please refer to the weaknesses

---

### Note · Authors · 2026-01-09

I have read and agree with the venue's withdrawal policy on behalf of myself and my co-authors.